# Sendai Virus and a Unified Model of Mononegavirus RNA Synthesis

**DOI:** 10.3390/v13122466

**Published:** 2021-12-09

**Authors:** Daniel Kolakofsky, Philippe Le Mercier, Machiko Nishio, Martin Blackledge, Thibaut Crépin, Rob W. H. Ruigrok

**Affiliations:** 1Department of Microbiology and Molecular Medicine, Faculty of Medicine, Medical School, University of Geneva, 1211 Geneva, Switzerland; 2Swiss-Prot Group, Swiss Institute of Bioinformatics, School of Medicine, University of Geneva, 1211 Geneva, Switzerland; philippe.lemercier@sib.swiss; 3Department of Microbiology, School of Medicine, Wakayama Medical University, Wakayama 641-8509, Japan; mnishio@wakayama-med.ac.jp; 4Institut de Biologie Structurale (IBS), CEA, CNRS, Université Grenoble Alpes, 38058 Grenoble, France; martin.blackledge@ibs.fr (M.B.); thibaut.crepin@ibs.fr (T.C.)

**Keywords:** Sendai virus, mononegavirus, RNA synthesis

## Abstract

Vesicular stomatitis virus (VSV), the founding member of the mononegavirus order (*Mononegavirales*), was found to be a negative strand RNA virus in the 1960s, and since then the number of such viruses has continually increased with no end in sight. Sendai virus (SeV) was noted soon afterwards due to an outbreak of newborn pneumonitis in Japan whose putative agent was passed in mice, and nowadays this mouse virus is mainly the bane of animal houses and immunologists. However, SeV was important in the study of this class of viruses because, like flu, it grows to high titers in embryonated chicken eggs, facilitating the biochemical characterization of its infection and that of its nucleocapsid, which is very close to that of measles virus (MeV). This review and opinion piece follow SeV as more is known about how various mononegaviruses express their genetic information and carry out their RNA synthesis, and proposes a unified model based on what all MNV have in common.

## 1. Introduction

The mononegavirus (MNV) order contains several families, including the *Rhabdoviridae* (rabies virus and vesicular stomatitis virus (VSV)), *Paramyxoviridae* (SeV and MeV), *Pneumoviridae* (respiratory syncytial virus (RSV) and human metapneumovirus (hMPV)) and *Filoviridae* (Ebola and Marburg viruses). Their non-segmented negative-strand RNA genomes are 11 to 19 kb in length with 5 to 10 tandemly arranged genes containing gene start (GS) and gene end (GE) signals that direct the synthesis of individual capped and polyadenylated mRNAs. Some genes can express more than one protein due to the presence of overlapping open reading frames via alternate ribosomal start codons and, perhaps uniquely for MNV, via co-transcriptional mRNA editing. This central protein-coding region is flanked by the replication promoters (or parts thereof) at the genome’s very 3′- and 5′-ends, termed 3′-leader and 5′-trailer, the latter being the complement of the 3′ end of the antigenome (Figure 1).

## 2. Mononegaviruses

### 2.1. MNV Nucleocapsids

MNV genomes are never found as free RNA, but always enclosed within a non-covalent homopolymer of the viral nucleoprotein (N). This assembly of ssRNA and multiple copies of N, the nucleocapsid (NC), can be considered as the true MNV genome as only genome RNA within this structure is normally recognized by the viral polymerase (pol). MNV N cores are composed of two lobes and a positively charged groove in between where the RNA is bound. Each N subunit also has amino- and carboxy-terminal arms that dock with binding sites on adjacent protomers (Figure 2), forming a stable coat of protomers along the genomic RNAs [1,2,3,4,5,6]. This system of domain swapping forms a helical structure along the entire length of the genomic RNAs, and protects even the very ends of the RNA of SeV to RNase A digestion under conditions in which less than one-hundredth of this RNase reduces all the RNA within ribosomes to small fragments [7]. SeV nucleocapsids are also stable enough to high salt and pressure that they can be isolated in pure and soluble form by equilibrium sedimentation in cesium chloride density gradients, and remain active for RNA synthesis when supplemented with extracts of transfected cells [8]. Given the robustness of this structure, it is unlikely that the N homopolymer ever dis-assembles, even when the protomers are separated from the genomic RNA during RNA synthesis.

In contrast to positive-strand RNA viruses, negative-strand RNA viruses do not have nor need helicases, as their genomic RNA is always bound to N that binds only to ssRNA. However, they do need to avoid the possible annealing of genomic and viral mRNAs, especially newly formed ones. MNV NC assembly thus presumably starts at or near the genomic 5′ ends as they emerge from pol during replication, and for MeV, its N binds more tightly to the nt sequence at its 5′ end than that at its 3′ end [9]. There is little detailed/direct experimental evidence of this process. However, when SeV RNA synthesis is carried out in cell extracts in which genome replication occurs, and the growth of the genome chain can be followed after its synchronous initiation, the nascent genome chains are found as progressively longer nucleocapsids [10]. This is evidence that genome synthesis and its assembly occur concurrently, but not necessarily that these two processes are coupled in the sense that one is dependent on the other (see below). These experiments found that the SeV genome was formed at roughly 1.7 nt/s, similar to the rate of VSV mRNA synthesis in vitro and MeV RNA synthesis in vivo, at 3 nt/s [11,12]. Compared to the rate of bacterial and yeast RNA synthesis at 40–80 nt/s, and RNA pol II at several hundred nt/s, this snail’s pace of MNV RNA synthesis presumably reflects the fact that MNV template RNAs are buried within their NCs and need to be uncovered before they can be copied.

### 2.2. MNV RNA Synthesis

MNV genome NCs are templates for two forms of RNA synthesis, that of mRNAs (termed transcription) and that of genome replication via antigenome NCs. MNV polymerases are complexes of the large L protein, responsible for all its catalytic activities, and a P (phosphoprotein) co-factor that oligomerizes due to a centrally located domain (P^OD^) (Figure 2). The SeV P oligomer was first predicted to function as a trimer, but crystals of its oligomeric domain showed a parallel coiled-coil tetramer [13,14]. P proteins thus have both N- and C-ter “tails” (P^N/C-ter^) that, except possibly for their extremities, are intrinsically disordered, providing great flexibility for their multiple interaction domains [15,16,17,18].

Although MNV nucleocapsids never disassemble once formed (as far as we know), their N protomers must be transiently separated from the genome RNA for the latter to enter pol’s template channel that leads past the synthesis chamber in pol’s core (Figure 5). The transient displacement of the Ns during MeV RNA synthesis is thought to be due to the transient and multi-faceted association of P^C-ter^ with N of the assembled NC (N^NC^) [19,20,21]. This may induce the lobes of N^core^ to twist relative to each other as pol moves down the NC during RNA synthesis, converting these displaced N^NC^ from a closed to an open conformation in which RNA is no longer bound. This separation of N subunits from the template RNA and their replacement after the template has exited pol is presumably the rate-limiting step in MNV RNA synthesis.

### 2.3. 3′ end Promoters

Rhabdo- and pneumoviruses have monopartite 3′ end promoters, i.e., all critical sequence elements for the initiation of RNA synthesis are within one segment of the leader region, mostly towards its 3′ end. In contrast, paramyxo- and filoviruses have bipartite promoters; the 3′-most promoter element (PE1) within leader, and PE2 within the 5′ UTR of the invariably first N gene (Figure 1). PE1 and PE2 are separated by a spacer region that includes the N mRNA start site (sometimes referred to as TSS, transcription start site) near position 56 from the genome 3′ end. Initiation of vRNA synthesis from this end not only requires both elements; the distance between them is critical. The separation of PE1 and PE2 can sometimes be altered without serious loss of activity, but only if this change is of hexamer length (+/− 6nt). Insertion or deletion of even a single nt in the spacer region, or a stretch of RNA of non-hexamer length, will inactivate these promoters [22,23,24,25]. Both paramyxo- and filovirus bi-partite promoters are thus governed by the “rule of six” [26].

All aspects of paramyxovirus RNA synthesis are governed by the “rule of six”. All these genomes found in nature are a multiple of 6 nt long, and only minigenomes of hexamer length replicate well in cell culture. This rule imposes a hexamer phase on the entire genome, which is composed of a series of hexa-nt bound to each N subunit. Paramyxovirus PE2 simply consists of three contiguous hexamers where only one or two of these nt are at all important, and for SeV and PIV5 these nt are found in hexamer positions whose nt bases point towards the solvent [27]. Because paramyxovirus NCs contain approximately 13 subunits/turn [1,4,6,28,29], these tripartite PE2s are juxtaposed on the same axial face of the NC helix as the 3′ end of the genome, presumably for concerted recognition by the viral polymerase (Figure 3).

The manner in which this bipartite promoter operates was examined for PIV2, a close relative of PIV5 and mumps virus. Ten residues of their N RNA-binding groove contact the RNA, nine with the ribose-PO4 backbone, and gln202 contacts a nt base [1]. The RNA-binding groove of MeV is virtually identical to that of PIV5 [28], and the two are compared in Figure 4. Minigenomes containing wild-type N/Q202 require both PE1 and PE2, as well as hexamer length for activity (i.e., wt hexamer phasing). However, when Q202 is mutated to one of several other residues, PE2 is no longer required at all, and non-hexamer length minigenomes remain active [30]. PE1 was proposed to contain a negative element, namely Q202 contacting the genomic 3′ end uridine where all vRNA synthesis likely begins (Figure 4). This interaction presumably prevents promoter activity unless pol can also simultaneously interact with the correctly phased PE2 tripartite repeat [31]. In this manner, paramyxovirus bipartite promoters ensure the hexamer phase of the entire genome, including that of the cis-acting mRNA editing signal where hexamer phase participates in regulating this process [32].

Filoviruses share many properties with paramyxoviruses, e.g., Ebola virus (EBOV) also employs mRNA editing, in this case to express its full-length receptor-binding protein, although there are also important differences. Bipartite promoters appear to be associated with MNV families that co-transcriptionally edit their RNAs [35]. Recently, the rule of six was found to also govern EBOV mRNA synthesis [36]. As bipartite promoters govern the initiation of RNA synthesis from the genome 3′ end, the obvious conclusion is that the EBOV transcriptase, like its replicase, must engage the template at the genome’s very 3′ end. The EBOV transcriptase would then arrive at TSS either by synthesizing leader RNA or by scanning the template (see below). An EBOV leader RNA has not been reported [37], but leader RNAs in infected cells tend to be unstable, and are best seen when transcription is carried out in vitro. For example, a prominent 47 nt-long leader RNA is the initial product of VSV mRNA synthesis in vitro [38].

### 2.4. Leader Regions and the Control of RNA Synthesis

Both plus-strand leader and minus-strand trailer RNAs (Figure 1) are found in SeV and VSV-infected cells, where the latter is thought to simply be the product of abortive genome synthesis (i.e., when the nascent trailer RNA is not concurrently assembled with N [39]). For VSV, the leader region was proposed to be central in determining whether its pol acts as a transcriptase or replicase; the choice was proposed to depend on whether or not the nascent leader RNA is assembled with N as its 5′ end exits pol [40].

Mechanistically, it is of course easier to separate the N chain from the genome RNA starting at a free 3’ end, than from TSS where both sides of the RNA chain are bound to nucleoprotein. For viruses with monopartite promoters such as VSV, their minigenome replication does not follow the rule of any integer. Here, dimethyl sulfate modification of nt bases within NCs, an indication of their accessibility to the solvent, occurs more readily and is independent of any integer phase, in contrast to that of SeV [32,41,42]. VSV pol can apparently recognize the monopartite promoter sequence within the NC, leading to the separation of the genome 3′ end from the N polymer, followed by its entry into L’s template channel and its alignment with the RdRp active site. For viruses with bipartite promoters, recognition of the PE2 promoter sequences within the NC may be a prerequisite for the RNA 3′ end to dissociate from the N chain, due the negative effects of PE1.

### 2.5. The RdRp Template Channel and Recognition of Cis-Acting Signals

Some MNV gene junctions are highly conserved, e.g., VSV, and SeV, whose intergenic regions (not copied into complementary RNA) are conserved and only 2–3 nt long. In contrast, the intergenic regions of other viruses are quite variable in length and sequence, even within different genera of the same family. In the latter cases, after terminating the upstream mRNA, transcriptases presumably scan the RNA within the NC and reinitiate mRNA synthesis upon encountering the next GS. Even for VSV and SeV, the 2–3 nt intergenic regions can be expanded by at least 200 nt without abrogating the ability of pol to specifically reinitiate mRNA synthesis when the ectopic GS is encountered. These studies have more accurately defined these GS, to be composed of at least the tail-end of the upstream GE sequence, the 2–3 nt intergenic region, as well as ca. the first 10 nt of gene start [43,44], all of which can be contained within the 20–25 nt-long pol template channel (Figure 5).

High resolution structures of the VSV [45], PIV5 [46], HMPV [47], PIV3 [48] and RSV L proteins [49,50] are now available. Their RdRp modules/domains are similar to those of other RNA viruses, and can be modeled as a right hand with thumb, fingers and a palm domain that surround the synthesis chamber containing the conserved GDN motif and the divalent cations of the active site [51]. There are channels where the ssRNA template enters and exits the RdRp module as it passes the synthesis chamber. There is a further channel(s) from the synthesis chamber where the nascent product RNAs exit pol, and a further channel or pore where NTPs gain access to the active site. Another feature of these structures is a priming loop that is thought to stabilize the NTPs that form the initial phosphodiester bond. In some structures this loop points towards the synthesis chamber and in others points away, and this presumably indicates whether pol is in the initiation or elongation mode, respectively. This loop needs to move away for the RNA:RNA hybrid (of about 7–10 bp) to form and fill the active site cavity, which is essential for the register of nt addition during RNA synthesis. This change in pol conformation is, of course only one of several that must occur during RNA synthesis.

During MNV RNA synthesis, N^NC^ need to be transiently separated from the template RNA; the RNA-free N chain presumably traveling around L’s exterior and rejoining the template RNA as it exits pol (Figure 5), maintaining protomer phasing. When pol movement on the template is not coupled to nt addition, Brownian motion may be involved, even over large distances. However, SeV N and P proteins are hot spots for phosphorylation turnover, which could also provide energy for translocation. In either case, if a ratchet mechanism linked to nt addition is not in play, this would permit scanning upstream for gene junctions with overlapping GS/GE.

**Figure 5 viruses-13-02466-f005:**
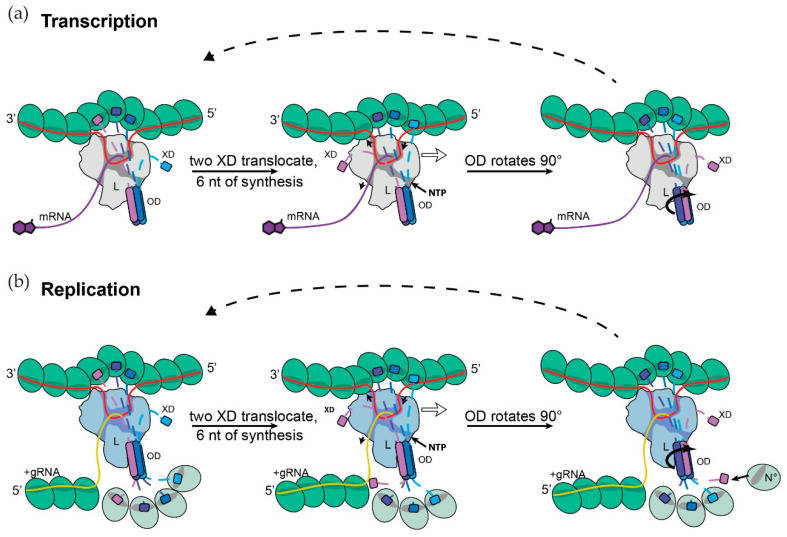
A cartwheeling model for SeV RNA synthesis. (**a**) Transcription. For SeV, the entire P^N-ter^ up to P^OD^ can be deleted without affecting transcription, the P^N-ter^ of P4-L appear not to have any role in this process [8,52], and are not shown for simplicity. The tetrameric P^OD^ is shown binding to RdRp domain of L (large irregular shape) with 2 of its 4 strands, and 3 of the 4 P^XD^ (helical bundle or X domain) in contact with N^NC^ that have been separated from the template RNA so that it can pass through L’s template channel. In this cartwheeling model, for each 6 nt added to the nascent mRNA chain, the free, leading (right-most) P^XD^ binds to the leading N^NC^ to separate it from the template RNA, the lagging P^XD^ is released from the lagging N^NC^ as this subunit rebinds the template RNA upon exiting pol and P^OD^ rotates by 90° so that it remains bound to L with 2 of its 4 strands. L’s channels are indicated by a darker shading, and the movement of the template RNA is indicated by arrowheads. (**b**) Replication. Same as (**a**) except that the 4 P^N-ter^, each associated with N^0^ (lighter color than N^NC^) are shown in addition. L is shaded differently to indicate its conformation may differ from that of the transcriptase. As P^OD^ rotates, the left-most N^0^ binds to the nascent replicate chain as it exits pol, and the vacant P^N-ter^ is recharged with N^0^ from a pool of P4-N^0^ (right-hand side). As long as there is sufficient P4-N^0^ in the cell to support genome replication, genome synthesis and its assembly into NC will be coupled in that they will occur concurrently.

When the scanning transcriptase encounters GS within the template channel, the initiating template pyrimidine will presumably align with the active site and the priming loop moves back in place to stabilize the initiating NTPs. Wandzik et al. have recently visualized the conformational dynamics of the influenza polymerase during the complete transcription cycle, focusing on template trajectory [53]. The channel through which the template RNA travels is indeed narrow, with multiple residues that line the channel available to interact with the nt bases. Such interactions presumably account for how transcriptases recognize GS or editing sites wherever they are located. Interaction of these channel residues can presumably also modify the nucleotide addition cycle, not only during formation of their mRNA polyA tails (by pol stuttering in response to GE [54]), but also during the editing of their mRNAs. A single such cycle starts with the template at the so-called +1 position and the complementary incoming NTP selected by Watson–Crick rules. Upon binding the incoming NTP, a critical methionine of motif F stacks on the nt base, inducing the closure of the active site with both coordinated divalent cations in a configuration suitable for catalysis. The 3′-OH of the nascent strand nucleophilically attacks the α-PO4 of the incoming NTP, a new phosphodiester bond is formed, the active site reverts to the open position and pol translocates along the template to the next +1 position.

When SeV and bPIV3 pols encounter their P gene editing signal, (essentially 3′ UUUUUUCCC) during mRNA synthesis and the editing site (underlined) is at the +1 position, the nascent chain of the RNA hybrid can slip backward relative to the template strand by one position before GTP can be incorporated. GTP is then added opposite the upstream C, adding a single pseudo-templated G to the nascent mRNA and bringing the editing site back to the original +1 position. For SeV and bPIV3, this editing cycle occurs at a specific frequency (SeV adds only a single G to the mRNA at ca 40% frequency, whereas bPIV3 adds 1 to 6 Gs at roughly equal frequency) and this difference is determined in part by the nature of the two nt directly upstream of the editing site (3′UG for SeV and 3′ AA for bPIV3 [55]). Apparently, the manner in which the residues which line the template channel interact with these two upstream nt is one of the methods that determines the mRNA editing phenotype. Remarkably, simply altering the hexamer phase of the editing signal can also affect this change [32].

### 2.6. Coupling of Genome Synthesis and Nascent Chain Assembly during Replication

It has long been suggested that the balance of transcription and replication are part of a self-regulatory system responding to the amount of N available for genome assembly [12,39,56,57]. Monomeric, unassembled N needed for NC formation (N^0^) is present in the cell, such as L, also bound to P4, via their N-ter tails [13,58,59]. As N^0^ in P4-N^0^ is prevented from spontaneously assembling on irrelevant RNAs (by various mechanisms in different MNV, reviewed in [46]), N^0′^s self-assembly is thus restricted to that of genome replication. P and N^0^ localize to cytoplasmic liquid droplets [15,60], which ensures that their local concentrations remain high, favoring P4-N^0^ formation.

Structures of MNV L have identified five conserved domains: RNA-dependent RNA polymerase (RdRp), poly-ribonucleotidyltransferase (PRNTase that caps the mRNA 5′ ppp-end), connecting domain (CD), methyltransferase (MTase that further modifies the cap), and the C-terminal domain (CTD) [45,46,47,48,50]. Comparison of different P-L structures suggests that they represent pol at different stages of mRNA synthesis. For example, the priming loop of the PRNT domain first points towards the central cavity (or synthesis chamber) to initiate mRNA synthesis. This loop is then withdrawn and replaced by the PRNT intrusion loop with the catalytic HR motif needed for capping, and this is followed by the repositioning of the MTase-CTD module directly above the PRNTase domain, which positions the active site of the MTase for productive capping and methylation, as in the PIV5 structure [46].

When new L (devoid of P) is available (e.g., via translation), it can associate with either P4 or P4-N^0^ depending on their availability. It is unclear whether P4 and P4-N^0^ affect L structure differently, but they do condition whether L acts as a transcriptase or replicase. All vRNA synthesis starts at the genomic 3′ end with 5′ ppp-leader (or trailer) sequences. For SeV, pol that have read through the leader/N junction terminate soon afterwards, whereas those having initiated N mRNA synthesis go through to the end of the gene [57]. In the absence of the concurrent assembly of the leader RNA and its synthesis, leader RNAs apparently terminate spontaneously whether or not this occurs precisely at the leader/N junction. Leader RNA termination in turn is needed for pol to initiate at TSS and modify the N mRNA 5′ end. Capping the N mRNA 5′ end may determine whether this pol continues RNA synthesis as a transcriptase, responding to all cis-acting signals. When the leader RNA is concurrently assembled with N during its synthesis, leader RNA synthesis never terminates; the P4-N^0^/L replicase cannot, therefore, respond to cis-acting signals, and RNA synthesis only terminates when pol runs off the end of the template. In this scenario, termination of leader RNA synthesis is central to L acting as transcriptase. Notably, once fixed as a transcriptase or replicase, these pols cannot interconvert until either process has been completed. Transcriptases changing to replicases in mid-stream, or vice versa, would be highly counterproductive.

### 2.7. Model

All MNV polymerases require a P protein cofactor that form oligomers via the centrally located P^OD^, and all (with the exception of mumps virus [61]) form parallel structures whose C- and N-ter tails emanate from P^OD^ in the same direction. Even though rhabdovirus P oligomers are dimers, filovirus P oligomers are trimers, and paramyxo- and pneumovirus oligomers are tetramers, they would all presumably operate in the same manner, as they all carry out the same biochemistry of transcription and replication, and something as fundamental as how RNA synthesis is controlled would have been set very early in MNV evolution. Genetic evidence has accumulated that the P^OD^ of the tetramers binds to the surface of L, and the recent cryo-EM P4-L structures of PIV5, hPIV3, RSV and hMPV have shown this binding in detail [46,47,48,50]. All bind to the same region of RdRp domain of L, with one significant difference: In the pneumovirus P4-L structures, two strands of virtually the entire tetrameric P^OD^ forms extensive contacts with the surface of the RdRp domain of L. In PIV5 and hPIV3, only the very C-ter end of P^OD^ contacts the RdRp domain, via two of the four strands, and the remainder of the tetrameric P^OD^ points away from L.

P itself has no catalytic activity, and is thought to play a structural role in helping pol along the NC during RNA synthesis. L can progressively move down the NC guided by P in several ways, including using two P^C-ter^ to “walk” along the N subunits of the N chain, or using more than two P^C-ter^ to cartwheel on the N subunits of the NC. When SeV P was predicted to function as a coiled-coil oligomer, it was proposed that P acted by cartwheeling across this N^NC^ assembly as L synthesized complementary RNAs, simply because there are more than two P^C-ter^ tails. However, for P^C-ter^ to cartwheel on the NC, all of P would also rotate, presumably while remaining bound to L. The recent P4-L structures offer a possible solution to the dilemma of how P^OD^ can rotate yet remain bound to L. In addition, they suggest a way whereby genome synthesis and its assembly into NCs can be tightly coupled, as the delay in adding even two protomers to the naked replicate RNA as it exits pol would expose 12 or more nt to which viral mRNA could anneal.

Pneumovirus P4 is bound to L by its parallel tetrameric coiled-coil, with two strands anchored to L’s surface. During RNA synthesis, however, P^OD^ may be less rigidly anchored due to P^C-ter^ interactions that separate N^NC^ protomers from the RNA template. Moreover, by super-positioning the individual RdRp domains of the two RSV structures (PDB: 6PZK and 6UEN), minor shifts between the interface of the L:P complex were identified, and Cao et al. have suggested that this interface may adopt a larger degree of conformational rearrangements during RNA synthesis [49]. The pneumovirus coiled-coil may thus be able to rotate while remaining bound to L; each quarter turn accompanying each 7 nt incorporated, and this would present the same interface of the coil to L during rotation (Figure 5B). It would be even easier for PIV5 P^OD^ to rotate, each quarter turn accompanying each 6 nt incorporated. In both cases, P^OD^ of P4-L could then act as a rotating spindle, allowing its P^C-ter^ to cartwheel across the NC. The same principle is applicable for filovirus P trimers, where each 120° turn accompanying each 6 nt is incorporated, and for rhabdoviruses, each 180° turn accompanying each 9 nt is incorporated.

Assuming that P4-N^0^/L acts as a replicase and that its nascent chain synthesis and assembly are coupled (in that they normally occur concurrently), during its P^OD^ rotation the N^0^ bound to its P^N-ter^ are available for their transfer to the emerging nascent genome RNA (Figure 5B). Similar to P^C-ter^ cartwheeling in which P^C-ter^/N^NC^ contacts are constantly broken and reformed with alternate contacts, N^0^ is constantly being transferred from P^N-ter^ of the P4-N^0^/L replicase to the nascent NC, and these vacant P^N-ter^ being replenished from a pool of P4-N^0^. P^OD^ acting as a rotating spindle could thus provide a mechanism to tightly couple genome synthesis to its concurrent assembly. Tellingly, when artificial means are used to reduce the availability of N^0^ during infection, not only is the generation of full-length genomes highly compromised, but this simultaneously leads to an enhanced innate immune response, presumably because the nascent genome chain is not protected by N^0^ assembly in a timely fashion, i.e., genome synthesis and assembly are no longer coupled [62].

## 3. Final Remarks

The MNV RNA synthesis machine may be simple in composition (N, P and L) but nevertheless carries out a remarkable number of different operations during transcription and replication. Given our very limited info about the conformational transitions that accompany these processes, the above scheme is far from the only one imaginable, but may help in elucidating the experimental pathways ahead.

## Figures and Tables

**Figure 1 viruses-13-02466-f001:**
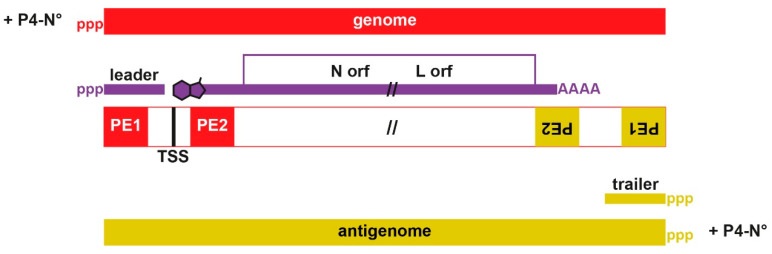
Schematic representation of bipartite promoters and the viral RNAs that ensue. The long rectangular box in the middle represents both the genome and antigenome. The positions of the bipartite promoter elements (PE1/PE2 in red for genome and dark yellow for antigenome) relative to the RNAs that ensue. Narrower purple and dark yellow lines indicate the free (unencapsidated) RNAs, and thicker lines indicate assembled NCs. The 5’ ends of the RNAs are marked with “ppp” and the cap group of the N mRNA with a schematic “m7Guanosine”. All the mRNAs are represented as a fusion of the N and L orfs; definitely not drawn to scale. TSS refers to transcription start site. P4-N^0^ refers to the complex between a tetrameric P and the nucleoprotein prior its binding to the viral RNA. After the binding of the nascent genomic RNA, the tetramer of P will leave the nascent N-RNA.

**Figure 2 viruses-13-02466-f002:**
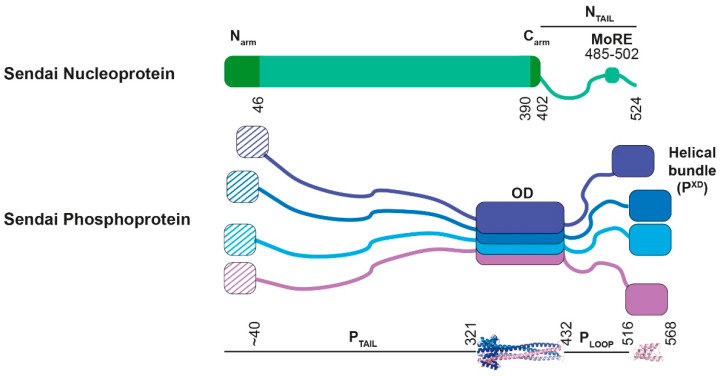
Schematic representation of the SeV nucleoprotein and phosphoprotein. Structured regions are shown as boxes, and intrinsically disordered regions as wavy lines. Numbers refer to their amino acid positions. MoRE stands for “molecular recognition element”. For the nucleoprotein, the numbers were taken from the paper of Zhang et al., [6]. The phosphoprotein forms tetramers. This was shown with the X-ray structure of the oligomeric domain (PDB 1EZJ) whereas the NMR structure of the helical bundle (PDB R4G) is monomeric. The first residues (1 to ~40) will fold into presumably two distinct α-helices that bind to N^0^, but the structure of N^0^P is not known.

**Figure 3 viruses-13-02466-f003:**
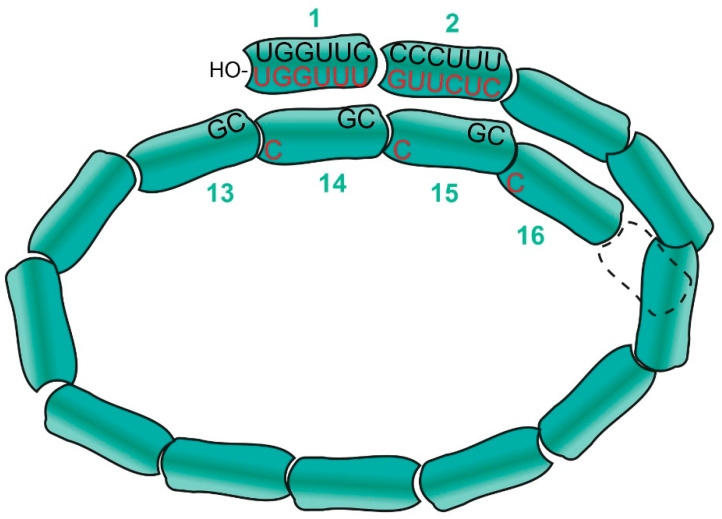
Model of the paramyxovirus NC helix with 13 subunits per turn. The N subunits are shown as quasi-rectangles numbered from the 3′-OH end, each binding precisely 6 nt. The relevant nt sequences of PE1 at the 3′ end and the tripartite PE2 in the N subunits on the next turn (and whose bases point towards the solvent) are shown for PIV5 (black) and for SeV (red).

**Figure 4 viruses-13-02466-f004:**
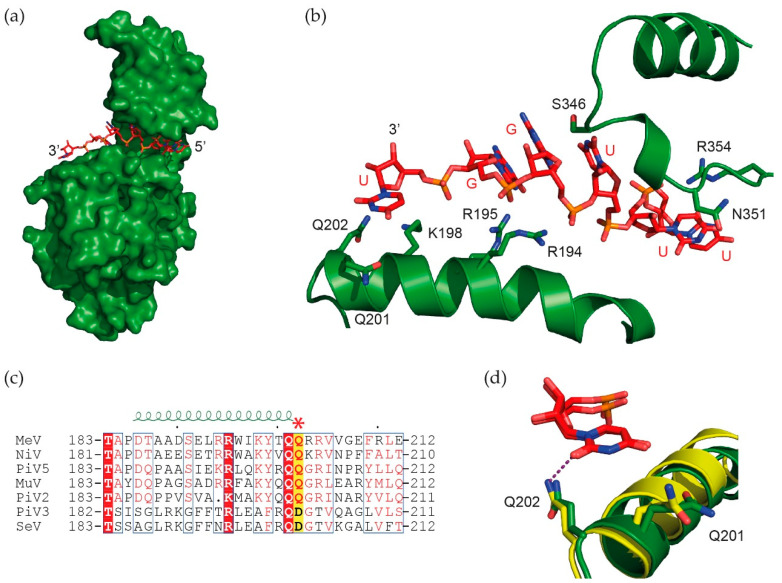
The binding of the critical glutamine Q202 of the N/RNA binding groove to the 3′ end of viral RNA. (**a**) Structure of the measles Ncore (PDB 6H5S) with the modeled RNA structure of the 3′ end; Ncore is shown in green and the RNA in red. (**b**) Zoom on the RNA binding groove in N showing the modeled RNA 3′ end (UGGUUU) interacting with residues Q202, K198, R195 and R194. (**c**) Sequence alignment of residues from 183 until 212 for nucleoproteins of several Paramyxoviruses. All of these proteins have 201-QQ-202, whereas Sendai and PIV3 have QD at the same position. (**d**) Detail of Q202 of measles (green) and PIV5 (yellow; PDB 4XJN) nucleoproteins binding to the last base of the viral RNA. For measles, the modeled RNA was the 3′ end of the viral RNA (UGGUUU) but the structure was from the cryoEM structure of N bound to AAAAAA. Thus, the position of the side chains of N could be slightly different. The figure was drawn using PyMOL [33] and ESPript [34].

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
