# Peer review of "Sendai Virus and a Unified Model of Mononegavirus RNA Synthesis"

_viruses, 2021, doi:10.3390/v13122466_

Round 1

Reviewer 1 Report

The manuscript, entitled “Sendai Virus and a Unified Model for Mononegavirus RNA Synthesis,” provides a comprehensive and up-to-date review of the mechanistic aspects of replication and transcription of negative-strand RNA viruses. The review is very detailed and has excellent illustrations. I think such reviews are very useful for those who are just entering the molecular virology and top professionals in this field. However, clarity in explaining some of the concepts and mechanisms of molecular functioning needs to be improved. Clarity is the key to reaching a wider readership and making a significant impact on the educational process. In the attached file, I have marked parts of the texts that require further work to improve understanding. I also gave some suggestions on how to do this. While my recommendations are not necessarily the most optimal ones, they can be considered a starting point for thought and perhaps help find the best way to express complex concepts in the authors' work to improve the text of this article.

Author Response

We are grateful to reviewer #1, who spent a good deal of time and effort pointing out that not everything in our text was as clear as it might have been, but there is a reason for this. This paper is not intended for those who do not have a firm grasp of this field, but for those who do and whose time in following the ever-increasing literature is limited, and who will appreciate our getting straight to the point of what is pertinent and needs to be said. If we tried to explain everything that is highlighted in color in detail, this paper would grow to the size of a chapter of Fields’ Virology, and might be useful to no one. We change a few errors found by reviewer#1.

Reviewer 2 Report

The review is very well written. The authors   explain  and summarize very complex molecular mechanisms concerning the regulation of genome transcription and replication of some Mononegaviruses with the help of figures.
This is an interesting article proposing a unified model  of Mononegaviruses RNA synthesis and I support its publication in its current form.  

 Minor points
-In figure 1, the description of  ‘+P4- N°’ is missing in the legend
-Line 106:  a space is missing after 2.3., and dot after '3’. ends’ should be deleted.

Author Response

Reviewer #2, as his/her review makes clear, had no problem in following the text. We have corrected his/her minor points in the legend for figure 1. However, the error in the text: “Line 106:  a space is missing after 2.3., and dot after '3’. ends’ should be deleted.” was not our error, someone has changed the text, the earlier version is this there: 2.3. 3’ end promoter. However, we have changed all the other errors reported by Reviewer #2.